# Kv10.1 Regulates Microtubule Dynamics during Mitosis

**DOI:** 10.3390/cancers12092409

**Published:** 2020-08-25

**Authors:** Naira Movsisyan, Luis A. Pardo

**Affiliations:** Oncophysiology Group, Max-Planck-Institute for Experimental Medicine, 37075 Göttingen, Germany; movsisyan@em.mpg.de

**Keywords:** calcium, Kv10.1, microtubule dynamics, ORAI1, spindle assembly checkpoint

## Abstract

Kv10.1 (potassium voltage-gated channel subfamily H member 1, known as EAG1 or Ether-à-go-go 1), is a voltage-gated potassium channel, prevailingly expressed in the central nervous system. The aberrant expression of Kv10.1 is detected in over 70% of all human tumor tissues and correlates with poorer prognosis. In peripheral tissues, Kv10.1 is expressed almost exclusively during the G2/M phase of the cell cycle and regulates its progression—downregulation of Kv10.1 extends the duration of the G2/M phase both in cancer and healthy cells. Here, using biochemical and imaging techniques, such as live-cell measurements of microtubule growth and of cytosolic calcium, we elucidate the mechanisms of Kv10.1-mediated regulation at the G2/M phase. We show that Kv10.1 has a dual effect on mitotic microtubule dynamics. Through the functional interaction with ORAI1 (calcium release-activated calcium channel protein 1), it modulates cytosolic calcium oscillations, thereby changing microtubule behavior. The inhibition of either Kv10.1 or ORAI1 stabilizes the microtubules. In contrast, the knockdown of Kv10.1 increases the dynamicity of mitotic microtubules, resulting in a stronger spindle assembly checkpoint, greater mitotic spindle angle, and a decrease in lagging chromosomes. Understanding of Kv10.1-mediated modulation of the microtubule architecture will help to comprehend how cancer tissue benefits from the presence of Kv10.1, and thereby increase the efficacy and safety of Kv10.1-directed therapeutic strategies.

## 1. Introduction

Kv10.1 (potassium voltage-gated channel subfamily H member 1, also known as Ether-à-go-go 1, EAG1; encoded by the *KCNH1* gene) is a voltage-gated potassium channel [1] almost exclusively expressed in the mammalian central nervous system [2], where it regulates neuronal excitability at high stimulus frequencies [3]. However, the functional conservation from cnidarians to humans [4] suggests that Kv10.1 may also serve other functions. Kv10.1 was among the early examples of potassium channels implicated in tumor progression. It increases the growth and aggressiveness of implanted tumors in mice [5]. Over 70% of solid human tumors are Kv10.1-positive, which correlates with an unfavorable prognosis [5,6,7,8,9,10,11,12,13,14]. In contrast, inhibition of the channel reduces tumor cell proliferation both in vitro and in vivo, making Kv10.1 a promising target for cancer therapy [15,16,17,18,19,20]. Nevertheless, the molecular mechanisms by which Kv10.1 favors cell proliferation and enhances tumor progression are poorly understood.

We have previously shown that, in non-neural cells, Kv10.1 localizes to the centrosomes and primary cilia [21], and that its expression occurs only during the G2/M phase of the cell cycle [22]. Downregulation of Kv10.1 induces accumulation of cells in the G2/M phase, suggesting that cells depleted of Kv10.1 require a longer time to complete G2/M, which implies the participation of the channel in the regulation of this phase of the cell cycle [22].

The successful completion of each step of the cell cycle is monitored by checkpoints, which block progression to the next stage until the quality criteria of the preceding one are met [23]. Two checkpoints function in G2 and M phases—DNA damage-induced checkpoint and spindle assembly checkpoint (SAC), respectively [23]. The DNA damage-induced checkpoint utilizes the ATM (ataxia telangiectasia mutated kinase)/ATR (ATM and Rad3 related kinase)—CHK2 (checkpoint kinase 2)/CHK1 (checkpoint kinase 1)—CDC25 (cell division cycle protein 25) axis, which configures the DNA damage response (DDR) machinery [24]. This checkpoint inhibits the complex cyclin B/CDK1 (cyclin-dependent kinase 1), precluding the initiation of mitosis and providing sufficient time for DNA repair [23].

SAC delays the onset of anaphase until all chromosomes are attached to the mitotic spindle in a bipolar fashion [25]. The biorientation of chromosomes occurs in a stochastic process—constant removal of incorrect kinetochore-microtubule connections and stabilization of those producing the required tension is the basis for achieving biorientation [25,26]. Hence, the process depends on the regulation of microtubule (MT) growth and shrinkage. The central effector of SAC is the mitotic checkpoint complex (MCC), which consists of MAD2 (mitotic arrest deficient 2), BUB3 (budding uninhibited by benzimidazoles 3 homolog), and BUBR1 (BUB1-related protein 1) [25]. MCC is recruited by kinetochores not attached to MTs and sequesters CDC20 (cell division cycle protein 20), hindering the activation of the anaphase-promoting complex/cyclosome (APC/C) [25]. Completion of the biorientation of chromosomes results in the inactivation of SAC, granting APC/C to initiate the anaphase.

The activation of kinases and phosphatases is not the only regulatory mechanism for progression through the cell cycle; changes in cytosolic ion composition are also essential for the process [27,28]. Transient changes in cytosolic [Ca^2+^] accompany the progression through different phases of the cell cycle [29,30,31], for example, during the metaphase-to-anaphase transition [32,33,34,35,36]. Moreover, Ca^2+^ ions can modulate the MT dynamics either directly or indirectly through changes in the protein make-up of MT-tips [37,38,39,40]. Therefore, the oscillations in cytosolic [Ca^2+^] during G2/M will induce changes in MT dynamics.

We designed this study to explore the status of the checkpoints regulating the G2 and M-phases in the absence of Kv10.1. Since it was recently proposed that Kv10.1 and the Ca^2+^ channel ORAI1 (calcium release-activated calcium channel protein 1) cooperate functionally to control Ca^2+^ influx [41], we studied whether Kv10.1 plays a role in providing the appropriate [Ca^2+^] for MT growth during spindle formation. We show that Kv10.1 downregulation leads to activation of SAC, preventing the formation of lagging chromosomes, commonly observed in cancer cells. The phenotype which we describe here partially depends on the functional interaction with ORAI1. Together with ORAI1, Kv10.1 changes the cytosolic Ca^2+^ behavior and, consequently, MT dynamics during mitosis. Besides, Kv10.1 influences the MT dynamics independently and opposite to its conducting properties, suggesting a dual effect of the channel on MT behavior.

## 2. Results

### 2.1. Downregulation of Kv10.1 Partially Activates the DNA Damage-Induced Checkpoint in the Absence of DNA Damage

Since Kv10.1 downregulation leads to a delay in the progression of cells through the G2/M phase [22], we investigated the status of DNA damage-induced checkpoint by assessing the abundance of DDR proteins.

HeLa cells were transfected with either control (scrambled, *Scr*-) or *KCNH1*-siRNA, synchronized at the G1/S border using a double thymidine block, and then allowed to proceed through the cell cycle. The cells were collected throughout the entire duration of the cycle and used to determine the abundance of DDR proteins by immunoblotting (Figure 1A–C). Analysis of the images revealed an increase in some of the DDR components in Kv10.1-deficient (KD) HeLa cells. The active phosphorylated forms of ATR (Ser42) [42,43] and CHK1 (Ser345, a phosphorylation site for ATR [44,45]) were more abundant in Kv10.1 KD cells (Figure 1A,B). The total level of phosphorylated (p-) ATR normalized for actin was doubled in KD cells (0.41 ± 0.03 a.u. vs. 0.20 ± 0.01 a.u. in control; Figure 1A). p-CHK1 was increased by 60% upon Kv10.1 knockdown (0.14 ± 0.03 a.u. vs. 0.09 ± 0.02 a.u. in control; Figure 1B). The total level of CHK2 was increased by approximately 70% in Kv10.1-deficient HeLa cells (0.24 ± 0.01 a.u. in the KD vs. 0.14 ± 0.01 a.u. in controls; Figure 1C). However, we observed no significant differences in the expression of phosphorylated BRCA1 (Ser1524—A phosphorylation site for ATR [46]), total CHK1, p21, and p-histone H2A.X (Ser139; Appendix A).

To elucidate whether the activation of components of the checkpoint actually responded to increased DNA damage, we studied the presence and extent of DNA lesions using single-cell gel electrophoresis, also known as the comet assay [47]. The analysis of DNA comets in asynchronously growing HeLa and hTERT RPE1 cells did not reveal any significant changes in the magnitude of DNA damage between the control and the Kv10.1 knockdown group (Figure 1D–H). The observed differences in the DNA damage-induced checkpoint components upon Kv10.1 depletion are hence independent of DNA damage.

### 2.2. Kv10.1 Downregulation Leads to Activation of Spindle Assembly Checkpoint and Decreased Occurrence of Lagging Chromosomes in HeLa Cells

Although the initiation of DDR occurs in the nucleus, some of the DNA damage-induced checkpoint proteins are also localized at centrosomes and are linked to SAC during mitosis, independently of DNA damage [48,49,50,51,52,53,54,55,56,57,58,59]. We, therefore, set out experiments to investigate a possible activation of SAC upon Kv10.1 knockdown. HeLa cells were transfected with *Scr*- or *KCNH1*-siRNA and synchronized at the G1/S border using a double thymidine block. Samples of synchronized cells were collected at the indicated time-points and analyzed by immunoblotting (Figure 2A–G). Densitometry analysis of the immunoblots showed a 20% increase in CDC20 in Kv10.1 KD cells (0.28 ± 0.05 a.u.) compared to the control group (0.23 ± 0.04 a.u.; Figure 2A). Similarly, MAD2 was 27% higher in Kv10.1 deficient HeLa cells (0.26 ± 0.02 a.u.) compared to the control cells (0.19 ± 0.02 a.u.; Figure 2B).

We also tested the status of the APC/C complex, a direct target of the MCC. The APC/C protein CDC27 (cell division cycle protein 27) was downregulated by 20% in Kv10.1 KD (0.24 ± 0.02 a.u.) with respect to the control cells (0.30 ± 0.03 a.u.; Figure 2C). Notably, the most significant difference between the groups occurred six hours after the release (a 60% decrease: 0.45 ± 0.09 a.u. in controls vs. 0.28 ± 0.05 a.u. in KD cells; Figure 2C).

The APC/C activity requires CDC27 [60], and the absence of CDC27 would originate a compensatory upregulation of other factors. The kinase WEE1 is a plausible candidate for such an upregulation. WEE1 inactivates the cyclin B1/CDK1 complex both at the entry into mitosis [61] and at the metaphase-to-anaphase transition [62]. WEE1 expression was increased by 30% in Kv10.1-deficient cells (0.16 ± 0.01 a.u.) compared to the control group (0.12 ± 0.01 a.u.; Figure 2D). Previously, we have shown that Kv10.1 downregulation in HeLa delays the degradation of cyclin A2 during G2/M phase [22]. Interestingly, cyclin A2 degradation has been suggested as a reporter for SAC activity—the higher the rate of MCC formation, the later cyclin A2 is degraded [63,64]. Additionally, in line with our previous observations [22], slightly broader peaks of CDC20 and WEE1 at the G2/M phase hint to extended and stronger SAC activity and the prevention of premature activation of APC/C in the absence of Kv10.1.

Besides MCC, other factors can reinforce the SAC signal. For instance, pVHL (von Hippel-Lindau protein) enhances SAC response by increasing MAD2 levels, which in turn reduces chromosome instability [64,65]. Immunoblot analysis showed a moderate increase in the total levels of pVHL in Kv10.1 KD cells (0.34 ± 0.03 a.u. vs. 0.27 ± 0.02 a.u. in control; Figure 2E). While MCC detects unattached kinetochores [25], the accuracy of already-formed kinetochore-MT attachments is modulated by the AURKB (Aurora kinase B)—PLK1 (Polo-like kinase 1)—MCAK (mitotic centromere-associated kinesin) axis [66]. We assessed the involvement of this pathway by monitoring changes in its expression during the cell cycle in synchronized HeLa cells transiently transfected with either *Scr*- or *KCNH1*-siRNA (Figure 2F,G). Interestingly, the total amount of AURKB was downregulated by 35% in Kv10.1-deficient HeLa cells (0.29 ± 0.03 a.u.) compared to control cells (0.40 ± 0.06 a.u.; Figure 2F). The localization of AURKB at the kinetochores is a crucial step in recruiting the SAC components [67] led by MAD2 [68]. Moreover, CHK1 and CHK2 help to correctly localize and activate AURKB, BUBR1, MAD1 at kinetochores [69,70,71,72]. Hence, although we observe a decrease in total AURKB in Kv10.1-deficient cells (Figure 2F), we can speculate that the amount of “effective” AURKB at the kinetochores may increase due to upregulation of both phosphorylated CHK1 (Figure 1B) and MAD2 (Figure 2B). Even a marginal loss of PLK1 activity impairs chromosome congression [66,73]. Like AURKB, PLK1 was 30% lower in Kv10.1 KD cells (0.11 ± 0.01 a.u. vs. 0.16 ± 0.02 a.u. in control cells; Figure 2G). However, the decreased total amount of PLK1 did not correlate with the enzymatic activity measured during the G2/M phase. A moderately higher PLK1 activity was detected both in Kv10.1-deficient HeLa (Figure 2H, left grey panel) and hTERT RPE1 cells (Figure 2H, right blue panel). As expected, cells synchronized at G1/S showed low overall PLK1 activity (Figure 2I). Thus, increased PLK1 enzymatic activity upon Kv10.1 loss in mitotic cells (Figure 2H) can contribute to the strength of SAC by modulating the kinetochore-MT attachments and facilitating the recruitment of MCC to kinetochores.

The upregulation of major components of SAC in synchronized HeLa cells depleted of Kv10.1 (summarized in Table 1) raises two questions. (i) Are SAC components upregulated due to chromosome misalignment during metaphase? And (ii) Does the more robust and prolonged SAC in Kv10.1 KD cells decrease the occurrence of lagging chromosomes? We approached these questions using HeLa cells transfected with either *Scr*- or *KCNH1*-siRNA again, and synchronized at the metaphase, to study chromosome alignment, and at anaphase—for the occurrence of lagging chromosomes. Bipolar mitotic spindle, kinetochores, and DNA were visualized in fixed synchronized cells using a confocal fluorescence microscope (Figure 2J,L). Kv10.1 knockdown had almost no effect on the number of defective metaphases, which are characterized by the presence of misaligned chromosomes or damaged DNA (Figure 2J,K). In contrast, *KCNH1*-siRNA-treated HeLa cells had approximately 20% less lagging chromosomes (29.32 ± 0.45%, than the control cells (35.78 ± 0.45%; Figure 2L,M), suggesting a more efficient error-correction of kinetochore-MT attachments in KD cells.

Altogether, the data indicate a moderate activation of SAC in the KD cells, sufficient to reduce the occurrence of lagging chromosomes.

### 2.3. Kv10.1 Knockdown Leads to Mitotic Spindle Tilting and Increased Microtubule Plus-End Growth

One of the causes of SAC activation can be an altered positioning of the mitotic spindle. HeLa cells with a larger tilting of the mitotic spindle require more time for biorientation of sister kinetochores due to prolonged activation of the spindle checkpoint [74]. Moreover, both PLK1 and pVHL, altered in Kv10.1-depleted cells, modulate the positioning of the mitotic spindle [75,76,77,78,79,80,81]. We measured the mitotic spindle angle in HeLa cells synchronized at metaphase (Figure 3A). The mitotic spindle angle was determined as the angle between the line connecting the two poles and the growing substrate (Figure 3B). As predicted, the mitotic spindle angle almost doubled in the Kv10.1 KD HeLa cells (17.20°) compared to the controls (9.68°; Figure 3C).

To understand the exact mechanism underlying the effect of Kv10.1 expression on SAC activity and mitotic spindle positioning, we studied the behavior of MTs. It is well-established that both SAC and spindle orientation on one side and MT dynamics on the other are highly interdependent [26,82,83]. We studied the effect of Kv10.1 KD on MT dynamics in cells treated with DME (dimethylenastron). DME is an Eg5 kinesin inhibitor that induces the formation of monopolar prometaphase spindles without affecting the assembly rates of MT plus-tips [84]. Yet, cells with altered MT dynamics and mitotic spindle angle form asymmetric monopolar spindles [85,86]. HeLa cells were treated with 2.5 µM DME, and the mitotic spindles were inspected under a fluorescence microscope for the presence of asymmetric monopolar spindles (Figure 3D). Kv10.1-deficient HeLa cells displayed more asymmetric monopolar spindles (33%) than the control group (24.20%; Figure 3E). The DME-assay has also been suggested as a screen-test for an altered mitotic spindle angle [86]. The DME-assay confirms the results from the direct measurement of the mitotic angle in metaphase-arrested cells: Kv10.1 downregulation is associated with an increased mitotic spindle angle.

To further validate the initial observation of increased MT dynamics in Kv10.1 KD cells, we measured the MT plus-end growth rates directly in living cells with transient Kv10.1 downregulation (upper blue panel, Figure 3F–I; Appendix A) or upregulation (lower green panel, Figure 3F–I; Appendix A). The growing MTs were detected by expressing EB3-tdTomato (end-binding protein 3 tandem dimer tomato fluorescent protein) and monitored in cells synchronized in prometaphase. The MT plus-end growth speed, growth length, and dynamicity were then computed with the U-Track software. It should be noted that we excluded the visibly asymmetric monopolar spindles from the analysis to ensure an unbiased assessment of MT dynamics. Due to that, however, cells with the most substantial effects are discarded, and our analysis display only minor changes in MT dynamics under the described conditions. Nevertheless, the growth speed was increased by 8.5% in Kv10.1 KD HeLa cells (20.77 µm·min^−1^ vs. 19.12 µm·min^−1^ in control cells; upper blue panel, Figure 3G). Similar to the growth speed, the frequency rate of switching from growth to shrinkage and vice-versa, known as dynamicity, was also increased in knockdown cells (top blue panel, Figure 3H). The displacement of the growing tip (growth length), was increased only by 3.6% upon Kv10.1 downregulation (upper blue panel, Figure 3I). Equivalent results were obtained in hTERT RPE1 cells with transient downregulation of Kv10.1 (Appendix A).

Opposite to Kv10.1 downregulation, overexpression of the channel globally reduced the MT dynamics (lower green panel, Figure 3G–I; Appendix A). The growth speed, dynamicity, and growth length were also lower in pKCNH1-WT-expressing cells compared to the cells transfected with the empty vector. This result strongly suggests that the alteration of MT dynamics is directly dependent on Kv10.1.

Despite the negative correlation between the MT dynamics and Kv10.1 expression, the causal relationship between Kv10.1 presence and the observed phenotypes remains to be defined. Whether the changes in the expression levels of regulatory proteins appear first, altering the MT behavior, or if the Kv10.1-dependent changes in MT dynamicity lead to mitotic spindle tilting and subsequently to SAC activation remains to be elucidated. We, nevertheless, favor the possibility that Kv10.1 participates in G2/M regulation by modulating the MT behavior, since spindle orientation is not affected by SAC inhibition [74].

### 2.4. Block of Kv10.1 or ORAI1 Currents Reduces Microtubule Dynamics

The absence of Kv10.1 abolishes both its canonical function (flow of K^+^) and any possible non-canonical function, such as its participation in supramolecular complexes. K^+^ currents hyperpolarize the membrane and, therefore, increase the driving force for Ca^2+^ [87], which controls MT dynamicity both in vitro and in vivo [37,38,88,89,90]. Ca^2+^ ions also regulate many MT-associated proteins present on the growing tips [40,91]. Besides, there is evidence that Kv10.1 interacts functionally with the Ca^2+^ channel ORAI1 and participates in store-independent Ca^2+^ entry (SICE) [41,92]. Thus, we tested the possible implication of such a functional tandem in MT growth during the G2/M phase.

We first studied the colocalization of Kv10.1 and ORAI1 in mitotic cells using proximity ligation assay (PLA). As shown in Figure 4A, using Kv10.1 and ORAI1-directed antibodies, the PLA gave rise to a signal, indicating the proximity of the two channels also in mitotic cells (zoomed-in inset, Figure 4A).

Further, we studied the effect of each of the channel conductance on MT dynamics in live-cell imaging experiments, as before. To block Kv10.1, we used the only specific blocker available, the monoclonal antibody mAb56 [15]. For ORAI1 inhibition, we used AnCoA4, a potent drug directed against the C-terminus of ORAI1 [93]. As controls, we used either the vehicle for AnCoA4 DMSO (dimethyl sulfoxide) or non-specific mouse immunoglobulin of the same isotype as mAb56—IgG (immunoglobulin G) κ2b. The results are presented in Figure 4B–E and Appendix A. Analysis of the time-lapse series showed that both AncoA4 and mAb56 significantly reduced all MT behavioral parameters: growth speed, dynamicity, and growth length (left grey and middle turquoise panels, Figure 4C–E). The effects were not additive: mAb56 and AnCoA4 combination treatment resulted in a smaller reduction in growth speed (right purple panel, Figure 4C). MT dynamicity was reduced by 10.7% in the cells exposed to AnCoA4 (left grey panel, Figure 4D), and by 8.7% in cells treated with mAb56 (middle turquoise panel, Figure 4D). Simultaneous treatment of the cells with AnCoA4 and mAb56 decreased MT dynamicity slightly less than each blocker alone (right purple panel, Figure 4D). The effect was similar on growth length, which was shortened by 14.1% in AnCoA4 (left grey panel, Figure 4E), 4.5% in mAb56 (middle turquoise panel, Figure 4E), and 9% in the combination (right purple panel, Figure 4E).

In summary, AnCoA4 produced significant reductions of all parameters of MT dynamics, and mAb56 still induced significant but smaller effects than AnCoA4. The combined treatment showed no additivity, suggesting a shared mechanism for both blockers and, therefore, a common pathway influenced by Kv10.1 and ORAI1. This observation would be compatible with MT growth regulation by Ca^2+^ entering the cell through ORAI1 in a Kv10.1-dependent manner. It is essential to point out that the changes induced by channel blockade are opposite to those caused by channel knockdown. Possible reasons for this difference are discussed below.

### 2.5. Kv10.1 Regulates Intracellular [Ca^2+^] Oscillations

We have shown that Kv10.1 and ORAI1 are in proximity, and that block of currents of either of them results in altered MT behavior during prometaphase, raising the possibility that Kv10.1 activity induces changes in cytosolic [Ca^2+^] through the functional interaction with ORAI1.

We tested the effect of blockade of ORAI1 or Kv10.1, and knockdown of Kv10.1, on the behavior of intracellular [Ca^2+^] during the G2/M transition in HeLa cells. As above, ORAI1 block was achieved by AnCoA4 treatment (using DMSO as a control); Kv10.1 was blocked by mAb56 (mouse IgG κ2b as a control); and Kv10.1 was transiently downregulated by *KCNH1*-siRNA (non-targeting *Scr*-siRNA—control). Before imaging, the cells were loaded with the intracellular Ca^2+^ indicator Fluo-4AM, and the fluorescent signal was recorded over 120 min, which roughly corresponds to the entire duration of M-phase (Figure 5A, Appendix A). The recorded signal was then normalized for photobleaching (Figure 5B). The area under the curve (AUC) and the width and the height of local maxima in the spectrum were determined, as shown in Figure 5C. The mean AUC, which indicates the relative intracellular Ca^2+^ concentration, was 16.7% smaller in AnCoA4-treated cells (left grey panel, Figure 5D). The block of Kv10.1 currents also reduced the relative amount of intracellular Ca^2+^, although by only 3.8% (middle turquoise panel, Figure 5D). In comparison, the average intracellular [Ca^2+^] was elevated by 10% in Kv10.1 KD cells (right blue panel, Figure 5D). Changes both in width (Figure 5E) and in the prominence of the peaks (Figure 5F) can explain differences in AUCs. The block of either of the channels resulted in narrower peaks (left grey and middle turquoise panels, Figure 5E). In contrast, the height of the peaks was not significantly affected (left grey and middle turquoise panels, Figure 5F). Opposingly, Kv10.1 silencing produced more prominent spikes (right blue panel, Figure 5F), but did not change the duration of the peaks (right blue panel, Figure 5G).

Calcium signals are not only encoded in the absolute magnitude of changes but also the frequency and amplitude of calcium oscillations [94]. Therefore, using the same normalized recordings, we calculated the changes in the frequency of intracellular Ca^2+^ oscillations converting the time domain of the signal into the frequency domain by Fast Fourier transform (Figure 5G). The block of ORAI1 significantly suppressed calcium oscillations. The absolute frequencies were already smaller in AnCoA4-treated cells (Figure 5H). Considering the relative power of each frequency, the average frequency of Ca^2+^ oscillations was 62.5% lower in ORAI1 blocked cells (3.45 mHz) than in the DMSO-treated cells (9.19 mHz; left grey panel, Figure 5I). As shown in Figure 5H,I (middle turquoise panel), the antibody-mediated block of Kv10.1 did not affect the frequency of oscillations, while Kv10.1 downregulation increased the average frequency of Ca^2+^ oscillations (right blue panel, Figure 5H). The weighted average frequency was higher in Kv10.1 KD (6.86 mHz) compared to the control group (6.32 mHz; right blue panel, Figure 5I).

As already mentioned, MT growth is affected by Ca^2+^ ions: Tubulin polymerization is inhibited in the presence of high [Ca^2+^] in vitro [37,38]. Therefore, Ca^2+^ oscillations would induce transitions from MT growth to MT shrinkage. A decrease of Ca^2+^ entry and oscillations would result in reduced dynamicity, while faster oscillations would accelerate the switching between MT growth and shrinkage. The block of ORAI1 decreases both the total cytosolic [Ca^2+^] and the frequency of cytosolic Ca^2+^ oscillations. The block of Kv10.1 has a similar effect, albeit less intense. The block of either of the channels shortens the duration of Ca^2+^ peaks, and thereby the absolute amount of Ca^2+^ ions entering the cell.

In contrast, siRNA-mediated downregulation of Kv10.1 has opposite effects on intracellular Ca^2+^, substantially increasing intracellular [Ca^2+^] and frequency of Ca^2+^ oscillations. Therefore, Kv10.1 current is required for regulating the duration of Ca^2+^ spikes, while the frequency is dependent not on the current but the physical presence of Kv10.1. Thus, our data strongly suggest that the origin of changes in MT dynamics under the different treatments reported here correlate to the effects on intracellular Ca^2+^ dynamics. Additionally, the mitotic spindles of Kv10.1 and ORAI1-inhibited cells appeared smaller and deteriorated (Figure 4A), which is in good agreement with the previously described effect of Ca^2+^ chelation on the mitotic spindle [39].

Differences between the effect of functional blockade and knockdown of the target are known for other proteins, like AURKB [95]. Our current observation of opposite effects of functional inhibition and depletion of Kv10.1 on MT dynamics and intracellular Ca^2+^ oscillations is also reminiscent of our previous results on the role of Kv10.1 in primary cilium homeostasis [21]. While the potassium current is necessary for primary cilium disassembly, the overexpression of the channel prevents reciliation in a way independent of the canonical function of the channel, since a non-permeant mutant also produces the effect, as does the C-terminal cytoplasmic domain without the transmembrane segments (see Figure EV3 in [21]). Similarly, abolishing the ability of Kv10.1 to conduct ions through mutagenesis is not enough to completely inhibit its tumor-promoting activity [96]. Furthermore, a non-conducting Kv10.1 splice variant (E65, found in cancer cell lines), activates CDK1 when injected into Xenopus oocytes [97]. The depletion of Kv10.1 would, thus decrease both the full length and the spliced forms of the channel and prevent E65-mediated CDK1 activation. In contrast, this effect would not occur when the channel function is inhibited.

Since canonic store-operated Ca^2+^ entry is not active during mitosis [98], the possible source of Ca^2+^ in dividing cells is SICE. Both membrane expression and activity of ORAI1 are dependent on SPCA2 (Secretory Pathway Ca^2+^-ATPase) [99,100,101]. Recently, it has been shown that SPCA2 also enhances Kv10.1 membrane expression, suggesting a new model of SICE [92]. In this scenario, it is tempting to speculate that the supramolecular complex between SPCA2, ORAI1, and Kv10.1 participates in providing the Ca^2+^ required for MT growth. Kv10.1, which is inhibited by Ca^2+^-calmodulin [102,103], would provide the negative control segment of the trio, terminating the signal.

## 3. Materials and Methods

### 3.1. Cell Lines

HeLa cells (DSMZ ACC 57) were maintained in RPMI (Roswell Park Memorial Institute Medium) 1640 (Gibco, Thermo Fisher Scientific, Waltham, MA, USA) supplemented with 10% fetal calf serum (Biochrom, Berlin, Germany) as recommended by the supplier. hTERT-RPE1 (ATCC CRL 4000) were grown in DMEM (Dulbecco’s modified Eagle’s medium; Gibco, Thermo Fisher Scientific, Waltham, MA, USA) supplemented with 10% fetal calf serum and 0.01 mg/mL hygromycin. No antibiotics were used during the experiments. For the imaging experiments, the cells were plated onto fibronectin (Sigma-Aldrich, Munich, Germany)-coated coverslips (#1.5, Paul Marienfeld, Lauda-Königshofen, Germany) in six-well plates (Greiner Bio-One, Frickenhausen, Germany) or glass-bottom four-well μ-ibidi (#1.5, Ibidi, Gräfelfing, Germany) imaging plates.

### 3.2. Transfections

Transient downregulation of Kv10.1 in HeLa and hTERT RPE1 cells was carried out using Interferin (Polyplus-transfection, New York, NY, USA) transfection reagent according to the manufacturers’ instructions. Co-transfection of siRNA and pEB3-tdTomato (end-binding protein 3 tandem dimer tomato fluorescent protein; a gift from Erik Dent; plasmid #50708, Addgene, Watertown, MA, USA; [104]) was performed with jetPRIME transfection reagent (Polyplus-transfection, Polyplus-transfection, New York, NY, USA). In all cases, the final concentration of siRNAs was 30 nM, and the following sequences of siRNAs were used: *KCNH1*, 5′-TACAGCCATCTTGGTCCCTTA-3′ (HP Custom siRNA, Qiagen, Hilden, Germany); *Scr* (Silencer negative control siRNA #1, Ambion, Kaufungen, Germany).

The co-expression of the hEAG1-ptracer (pKCNH1-WT; [5]) and pEB3-tdTomato in HeLa and hTERT RPE1 cells was achieved using the jetPRIME transfection reagent following the manufacturer’s guidelines.

### 3.3. Cell Cycle Synchronization

Depending on the experiment, HeLa cells were synchronized at various stages of the cell cycle. When needed, 4 h before the synchronization, the cells were transfected with the corresponding siRNAs.

Changes in proteins of interest over time were studied in HeLa cells synchronized at the G1/S border by growing the cells for two periods of 16 h with excess thymidine (2 mM; Sigma-Aldrich, Munich, Germany) and 8 h release in-between. The cells blocked at G1/S were then transferred into fresh growth medium, and cell samples were collected every 2 h between 0 to 6 h and 10 to 14 h, and every hour between 6 to 10 h (G2/M phase). The samples were then used for immunoblot experiments.

For metaphase and anaphase synchronization, the cells were treated only once with 2 mM thymidine for 16 h, then rinsed twice with DPBS (Dulbecco’s phosphate-buffered saline; Gibco, Thermo Fisher Scientific, Waltham, MA, USA) and released in fresh growth medium for 4 h. Subsequently, the cells were exposed to 2 ng/mL nocodazole for 4 h and washed again with DPBS as before. The cells were harvested either at this stage (for PLK1 activity) or synchronized further. For the anaphase, the cells were released for 40 min into a standard growth medium. In the case of metaphase, the cells were further incubated with 15 µM MG132 for 2 h at 37 °C and 5% CO_2_.

Monopolar spindles were produced by treating asynchronously growing HeLa cells with 2.5 µM DME (dimethylenastron; Calbiochem, Sigma-Aldrich, Munich, Germany) for 4 h at 37 °C and 5% CO_2_. The cells were then either immediately fixed and used for immunofluorescence or imaged within two hours.

For intracellular calcium imaging, HeLa cells were synchronized at the G2/M border by a double thymidine protocol followed by a release into a fresh growth medium for 7.5 h.

### 3.4. Protein Extracts

Proteins were extracted from the synchronized cell pellets with a non-denaturing lysis buffer (1% Triton X-100, 50 mM Tris-HCl, 300 mM NaCl, 5 mM EDTA) containing phosphatase and protease inhibitor cocktail (Roche, Penzberg, Germany) for 30 min at room temperature (RT) in 1:3 ratio (volume-to-volume, v:v). Nuclear proteins were extracted with lysis buffer containing 1% (v:v) NP-40, 0.1% (weight-to-volume, w:v) Na-deoxycholate, 300 mM NaCl, 50 mM Tris-HCl, and phosphatase and protease inhibitors. The lysates were centrifuged at 18,000× *g* for 15 min at 4 °C, and the supernatant was recovered and stored at −80 °C. Protein concentration was determined with the BCA (bicinchoninic acid) assay calibrated with BSA (bovine serum albumin provided with the kit) according to the manufacturer’s recommendations. Protein samples (50 μg) were denatured by incubation at 70 °C for 10 min with the NuPAGE Reducing agent (1:10, v:v; Invitrogen, Carlsbad, CA, USA) and the NuPAGE LDS Sample buffer (1:4, v:v; Invitrogen, Carlsbad, CA, USA).

### 3.5. SDS-PAGE and Immunoblotting

Low and mid-range molecular weight proteins were resolved in 4–12% NuPAGE Novex Bis-Tris Mini Gels (Invitrogen, Carlsbad, CA, USA) at 200 V in NuPAGE MOPS electrophoresis buffer (Invitrogen, Carlsbad, CA, USA) and blotted onto 0.2 µm pore size nitrocellulose membrane (Invitrogen, Carlsbad, CA, USA) at 50 V for 2 h at RT in 10 mM NaHCO_3_, 3 mM Na_2_CO_3_, 0.01% SDS (w:v), and 20% methanol (v:v) transfer buffer. High molecular weight proteins were separated in 3–8% NuPAGE Novex Tris-acetate Gels (Invitrogen, Carlsbad, CA, USA) in Tris-acetate electrophoresis buffer (Invitrogen, Carlsbad, CA, USA) at 150 V, and transferred onto a nitrocellulose membrane (0.45 µm, GE Healthcare, Chicago, IL, USA) by 10 mM NaHCO_3_, 3 mM Na_2_CO_3_, 0.01% SDS, and 15% methanol transfer buffer at 35 V and 4 °C, overnight. After transfer, the membranes were rinsed with distilled water and treated with Western Blot Signal Enhancer (Thermo Fisher Scientific, Waltham, MA, USA) following the manufacturer’s instructions. Non-specific binding was blocked for 90 min with either 0.1% casein (Roche, Penzberg, Germany) or 5% BSA (Sigma-Aldrich, Munich, Germany) in TBS (Tris-buffered saline, 20 mM Tris, 150 mM NaCl) with 0.1% Tween 20. The membranes were then incubated with the appropriate primary antibodies (Table 2) in blocking buffer overnight at 4 °C. After washing with TBST, the membranes were exposed to HRP (horseradish peroxidase)-coupled secondary antibody (Table 2) in blocking buffer for 45 min. Proteins were detected by incubating with a chemiluminescent HRP substrate (Millipore, Billerica, MA, USA) for 5 min and protected from light. The signal was detected with a ChemiDoc XRS system (Bio-Rad Laboratories, Feldkirchen, Germany) controlled by the Quantity One 1-D Analysis software v4.6.9 (Bio-Rad Laboratories, Hercules, CA, USA). Ten images were acquired between 1 s and 10 min of exposure or until saturation was reached. In some cases, the membranes were stripped using Restore Plus Western Blot stripping buffer (Thermo Scientific, Rockford, IL, USA) for 30 min and reused after blocking if no signal was detected in a 5 min exposure after incubating with HRP chemiluminescence substrate.

Changes in expression of the protein of interest were estimated by densitometry analysis using the ImageLab software version 4.1 (Bio-Rad Laboratories, Hercules, CA, USA) using the intensity profile of each lane and normalizing the volume intensity of the bands to the corresponding value for actin.

### 3.6. Immunoprecipitation

The scarce expression of Kv10.1 in HeLa and hTERT RPE1 cells made it necessary that the efficiency of knockdown was tested by immunoprecipitation with an anti-Kv10.1 monoclonal antibody (Kv10.1-33.mAb), and recovery with protein G magnetic beads (New England BioLabs, Frankfurt am Main, Germany) following the manufacturer’s instructions (Appendix A). The beads were washed three times with 500 µL of cold immunoprecipitation buffer (0.1% Triton X-100, 50 mM Tris-HCl, 300 mM NaCl, 5 mM EDTA and protease inhibitor cocktail) and Kv10.1 was eluted at 70 °C for 10 min in NuPAGE LDS Sample buffer (1:4, v:v).

### 3.7. Single-Cell Gel Electrophoresis

DNA lesions were detected by alkaline single-cell gel electrophoresis (comet assay) [105]. Cells were plated in a six-well plate, transfected with siRNA one day later and assayed after 48 h. All the steps were carried out under dim light and at 4 °C. The assay was performed in clean CometSlide 2 Well (Trevigen, Gaithersburg, MD, USA) slides coated with 0.5% normal melting point agarose, on which 15,000 cells/well in 1% low melting point agarose were layered. The slides were incubated in lysis buffer (2.5 M NaCl; 0.1 M EDTA; 10 mM Tris, 10% DMSO, v:v; 1% Triton X-100, v:v; pH, 10) for 1 h, placed in cold electrophoresis buffer (1 mM Na_2_EDTA; 300 mM NaOH; pH > 13) and denatured for 20 min. The electrophoresis was carried out for 20 min at 13 V (0.6 V/cm across the platform) and 300 mA. The slides were neutralized (0.4 M Tris; pH, 7.5) for 10 min and then fixed in ethanol, air-dried overnight, and stained with SYBR Gold (Invitrogen, Carlsbad, CA, USA) following the manufacturer’s instructions. Ten positions in each well of the still-wet slides were imaged on a Nikon TiE-Andor spinning disk microscope (Oxford Instruments, Abingdon, UK) equipped with a PLAN APO VC Air 20× 0.75 NA objective (Nikon Instruments, Amsterdam, Netherlands). Images were acquired with an Andor DU-888 X-9984 camera controlled with the NIS-Elements AR software v.4.50 (Nikon Instruments, Amsterdam, Netherlands). Maximum intensity projections of z-stack images with 0.2 µm z-step size were analyzed using CASPLab open-source software [106].

### 3.8. Immunofluorescence

To visualize mitotic spindles and kinetochores, cells seeded on coverslips were fixed with 2% *p*-formaldehyde, 60 mM PIPES, 27 mM HEPES, 10 mM EGTA, 4 mM MgSO_4_, pH, 7.0 for 5 min at RT, followed by methanol at −20 °C for 5 min. After blocking for 90 min in 10% BSA in DPBST (0.1% Tween-20), the cells were incubated for 2 h at RT with anti-α-tubulin, anti- CENP-B (centromere protein B), or anti-γ-tubulin (see Table 3 for dilutions and specifications), followed by a one-hour incubation with the secondary antibodies conjugated to Alexa Fluor-488, -546 or -633, respectively. Nuclei were counterstained with Hoechst 33342 for 5 min at RT, washed three times with DPBST for 5 min at RT, and mounted using Prolong Gold antifade reagent (Invitrogen). The slides were imaged on a Leica TCS SP5 II confocal laser scanning microscope (Leica Microsystems, Wetzlar, Germany) equipped with HCX PL APO Oil 40× 1.25 NA objective. Images were acquired with a z-optical spacing of 0.2 µm using LAS AF (Leica Microsystems, Wetzlar, Germany) software, and processed with the image analysis software ImageJ2 [107,108,109] (U. S. National Institutes of Health, Bethesda, MD, USA) and Imaris (Bitplane, Oxford Instruments, Belfast, UK).

### 3.9. Determination of Fan-Shaped Mitotic Spindles

The occurrence of asymmetric (fan-shaped) mitotic spindles was determined in HeLa cells synchronized in prometaphase with DME 48 h post-siRNA transfection. Defective mitotic spindles were analyzed by immunofluorescence microscopy detecting microtubules and DNA. The occurrence of fan-shaped mitotic spindles was defined as a percentage of the defective spindles from the total number of monopolar mitotic spindles.

### 3.10. Determination of Lagging Chromosomes and Defective Metaphases

To detect lagging chromosomes in anaphase, HeLa cells were transfected with *Scr*- or *KCNH1*-siRNAs, and the synchronization with thymidine-nocodazole was started 4 h later. Cells were visualized by immunofluorescence microscopy detecting CENP-B-positive chromosomes. Only chromosomes that were both CENP-B-positive and separated from the two pole-oriented chromosome masses were counted as “lagging chromosomes”.

The defective metaphases were defined in the cells synchronized at metaphase and stained for α-tubulin, CENP-B, Hoechst 33342. Using Hoechst 33342 and CENP-B staining, the alignment of chromosomes was analyzed.

### 3.11. Determination of Mitotic Spindle Angle

Mitotic spindle angle and the distance between the poles were measured using Imaris v9.0 in HeLa cells synchronized at metaphase and stained for γ- and α-tubulin. Two spheres with a diameter of 1.5 µm around the maximum intensity of γ-tubulin were generated at the centrosomes. A third sphere was positioned as a projection of one of the centrosomes at the z-level of the opposite centrosome.

### 3.12. Determination of the Polo-Like Kinase 1 Enzymatic Activity

PLK1 activity was measured in lysates from cells synchronized at G1/S and prophase using CycLex Polo-like kinase 1 Assay/Inhibitor Screening Kit (MBL International Corporation, Woburn, MA, USA). The reaction was started by addition of 180 µL of Kinase Reaction Buffer to 20 µL cell lysates (50 µg protein) in microtiter wells coated with recombinant Protein-X as PLK1 substrate (CycLex Kit) and incubated for 2 h at 30 °C. The rest of the steps were performed according to the instructions in the kit protocol. Only lysis buffer was used as a negative control and 0.2 mU of purified recombinant PLK1—as a positive control. All samples were prepared in duplicates. The absorbance in each well was measured using dual wavelengths at 450/540 nm on a Wallac 1420 VICTOR2 microplate reader (Wallac Distribution GmbH, Freiburg, Germany).

### 3.13. Proximity Ligation Assay

The interaction of Kv10.1 and ORAI1 in asynchronously growing HeLa cells was assessed with the Duolink PLA kit (Sigma-Aldrich, St. Louis, MO, USA) following the manufacturer’s instructions using anti-Kv10.1 (mAb62; 1:2000, mouse monoclonal) and anti-ORAI1 (Alomone Biolabs; 1:200, rabbit polyclonal) antibodies and imaged with a Leica TCS SP5 II confocal microscope equipped with HCX PL APO Oil 40× 1.25 NA objective.

### 3.14. Live Cell Imaging

All time-lapse imaging experiments were performed on a Nikon TiE-Andor spinning disk confocal microscope equipped with a PLAN APO-TIRF 100× 1.49 NA oil immersion objective. Images were acquired with an Andor DU-888 X-9984 camera controlled with NIS-Elements software.

### 3.15. Assessment of Microtubule Dynamics

Microtubule plus-end assembly rates were determined by tracking EB3-tdTomato in living cells. HeLa and hTERT RPE1 cells were transfected with pEB3-tdTomato alone or in combination with siRNAs. 48 h later, the cells were treated with 2.5 µM DME in a phenol-red free medium for 3.5 h, and transferred to the environmental control chamber of the microscope (37 °C and 5% CO_2_) and further incubated until a total incubation time of 4 h. Throughout the entire experiment, the temperature was maintained constant at 37 °C. Images were acquired with 500 ms intervals for 1 min. EB3-tdTomato was excited using the 561 nm laser line, and the emission was collected with a 600/52 emission filter. The MT dynamics were analyzed with U-Track 2.2.0 following the authors’ guidelines [110,111,112].

### 3.16. Intracellular Calcium Imaging in HeLa Cells

Fluctuations in cytosolic [Ca^2+^] were monitored in HeLa cells synchronized at the G2/M phase using Fluo-4AM (Invitrogen, Carlsbad, CA, USA) [113]. After 6.5 h of release from the second block with thymidine, the cells were incubated with 10 µM Fluo-4AM in phenol-red free DMEM medium for 50 min, and then the medium was replaced, and image acquisition was performed at 37 °C and 5% CO_2_. Image acquisition started at 60 min after the addition of Fluo4-AM at five positions per well for 2 h every 10 s.

In each recorded image, ten cells were manually selected as regions of interest (ROIs; in total, 50 cells per condition), and three background ROIs of constant size were identified. The intensity values in each ROI were measured over time with the NIS-Elements software. The corrected total cell fluorescence (CTCF) for each ROI and time point was calculated as a difference of the cell integrated intensity and the product of multiplication of the area of the selected cell and the mean background fluorescence. The generated CTCF values were further analyzed using MATLAB. In the time domain, a linear function was fitted to the collected data to compensate for the baseline drift resulting from photobleaching. The baseline was then subtracted from the measured values, generating normalized CTCFs. Additionally, to make the collected data comparable between experiments, the values were expressed as a percent change to the respective time-point “0”. The area under the curve (AUC), prominence, and full width at half maximum (FWHM) were calculated using trapz and findpeaks functions. Further, the time domain was transformed into a frequency domain by applying a Fast Fourier Transformation. The two-sided, and then the single-sided spectra were computed. The frequencies were sorted after power spectrum density (PSD) in descending order, and the first 15 values were used to calculate the weighted average frequency, defined as a mean of the product of frequency and the respective PSD.

### 3.17. Statistical Analysis

All data were analyzed using Prism software v8.4.3 (GraphPad Software LLC, San Diego, CA, USA). The analysis used for each experiment is indicated in the corresponding Figure legend. Data are shown as the median with interquartile range or the mean with standard error (SEM). *p*-value < 0.05 was considered statistically significant.

## 4. Discussion

Our current understanding of the Kv10.1 implication in the G2/M phase involves both canonical (ion-channel activity) and non-canonical (massive intracellular domains and the splice variants present in the cell) mechanisms (Figure 6). On the one hand, Kv10.1/ORAI1 tandem triggers changes in intracellular Ca^2+^ behavior. The fluctuations in [Ca^2+^] can directly affect the MT growth and change the activity of many of the proteins, which can translate the information encoded in the frequency and amplitude of Ca^2+^ oscillations [94]. Considering the centrosomal localization of Kv10.1 [21], the decoding would take place at the centrosomes—a hub for a myriad of cell cycle regulators, including AURKA (Aurora kinase A) and PLK1 [114]. Ca^2+^/CaM transiently activates both these kinases during the G2/M transition [115,116]. It is, therefore, appealing to speculate that AURKA and PLK1 might react to Kv10.1/ORAI1 driven changes in cytosolic Ca^2+^ oscillations.

On the other hand, the physical presence of Kv10.1 both at the centrosomes and the split-forms in the cytosol hold a high potential of interacting and controlling the activity of many enzymes involved in mitosis. It is tempting to speculate that Kv10.1 can interfere with the activation of CDK1, AURKA, and PLK1 independently of its conductance. As already mentioned, the spliced form of the channel E65 activates CDK1 in Xenopus oocytes [97]. Interestingly, a positive feedback loop exists between AURKA, CDK1, and PLK1 [117,118,119,120,121]. All these three kinases are implicated in microtubule growth and SAC regulation [122,123,124]. Therefore, any imbalance in this system during mitosis can affect MT dynamicity, the positioning, or even formation of the mitotic spindle. In either way, SAC response will be altered, delaying, or speeding up the metaphase-to-anaphase transition.

## 5. Conclusions

The tumor specificity and ease of access of Kv10.1 make it an attractive candidate for cancer treatment. However, the clinical development of Kv10.1-directed therapies has faced difficulties due to the incomplete understanding of the underlying molecular mechanisms. Here, we attempted to bridge that gap and broaden our very limited knowledge on the implication of ion channels in cell growth. Furthermore, understanding the molecular mechanisms causing the different effects of functional inhibition or gene-silencing mediated downregulation of Kv10.1 can help to tailor cancer treatment to tumor phenotype. Therefore, although the current work in cell lines cannot be directly translated to the clinical scenario, we are persuaded that Kv10.1 will be a relevant oncological target in the future.

## Figures and Tables

**Figure 1 cancers-12-02409-f001:**
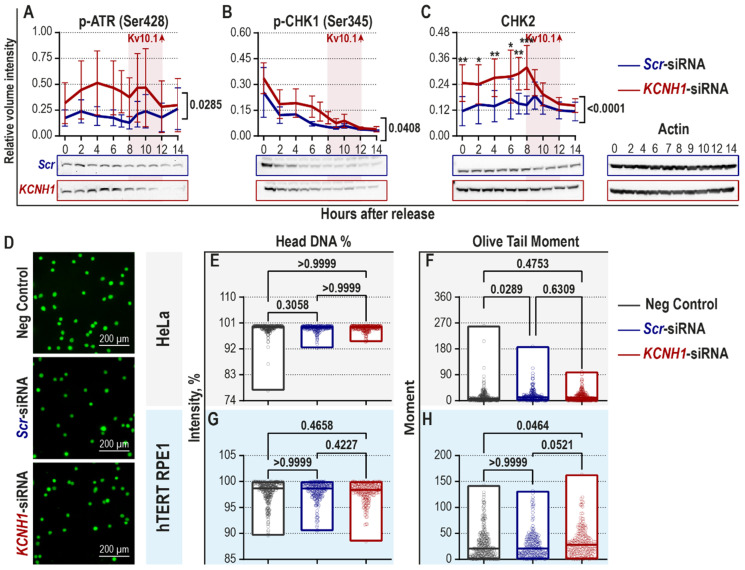
Kv10.1 downregulation activates the DNA damage-induced checkpoint in the absence of DNA damage in HeLa cells. (**A**–**C**) Hela cells were transiently transfected with either *Scr*- or *KCNH1*- siRNA, synchronized with double thymidine block at the G1/S border, and subsequently released from the block. Samples were collected at the indicated time-points after release from the block. Representative immunoblots and the results of densitometry analysis are plotted. The time interval corresponding to Kv10.1 expression is indicated in pink (*n* = 3 independent experiments; mean ± SEM; two-way ANOVA, repeated measures, Bonferroni post hoc test; the exact *p*-values indicate the significance of the global difference between the groups, and the *p*-value between single time points is indicated by asterisks: * *p* < 0.05, ** *p* < 0.01, *** *p* < 0.001). (**D**–**H**) HeLa and hTERT RPE1 cells were transiently transfected with either *Scr*- or *KCNH1*-siRNA, allowed to reach 70% confluency, harvested, and immediately used for single-cell gel electrophoresis (comet assay). (**D**) Representative images of the DNA comets stained with SYBR-Gold are given (max intensity projection, z-step, 0.2 µm, scale bar, 200 µm). (**E**–**H**) The comets were analyzed with the CASPLab software and Head DNA % and Olive Tail Moment are plotted in (**E**,**F**) for HeLa cells, and in (**G**,**H**) for hTERT RPE1 cells (*n* = 5 independent experiments with >750 comets per condition; Kruskal–Wallis test with Dunn’s post hoc test, min to max range with the median indicated as a line are shown, with *p*-values indicated for each pair). (Kv10.1—potassium voltage-gated channel subfamily H member 1; *KCNH1*—potassium voltage-gated channel subfamily H member 1 gene; *Scr*—scrambled; p—phosphorylated; ATR—ATM and Rad3 related kinase; CHK—checkpoint kinase; HeLa—Henrietta Lacks; hTERT RPE1—human telomerase reverse transcriptase immortalized retinal pigmented epithelial).

**Figure 2 cancers-12-02409-f002:**
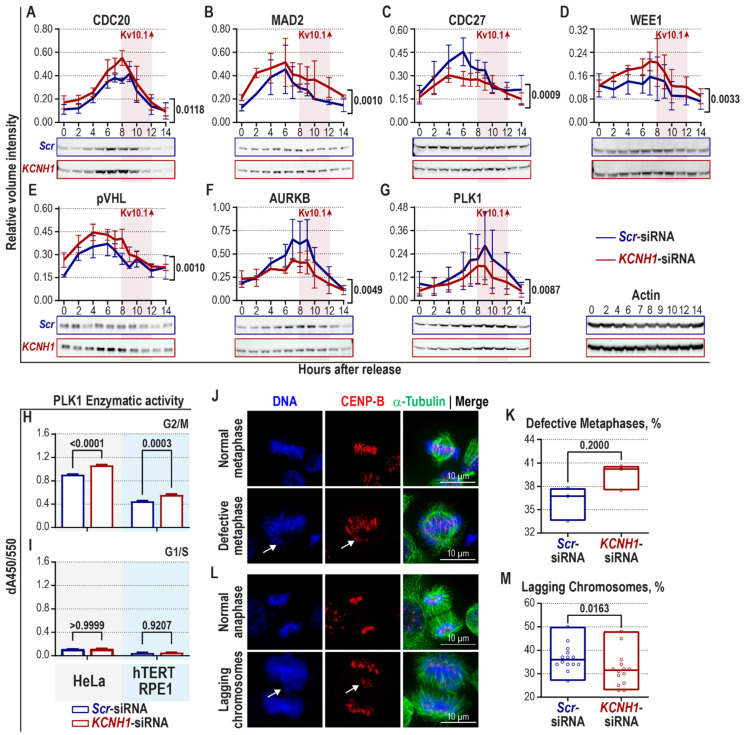
Activation of SAC in Kv10.1-depleted HeLa cells decreases the occurrence of lagging chromosomes. HeLa cells were transiently transfected with either *Scr*- or *KCNH1*-siRNA and synchronized with double thymidine block at the G1/S border (**A**–**I**) or with consecutive thymidine and nocodazole treatments at metaphase and anaphase (**J**–**M**). (**A**–**G**) Samples were collected at the indicated time-points after cell cycle re-initiation and immunoblotted. Representative immunoblots and densitometry analysis results are given. The time interval corresponding to Kv10.1 expression is highlighted in pink (*n* = 3 independent synchronization experiments; mean ± SEM, two-way ANOVA, repeated measures, Bonferroni post hoc test, the given *p*-value shows the significance of the global difference between the two groups. (**H**,**I**) PLK1 enzymatic activity was measured in whole-cell lysates from HeLa (left grey panel) and hTERT RPE1 (right blue panel) cells synchronized at G2/M (**H**) and G1/S (**I**) (*n* = 4 independent synchronization experiments; two-way ANOVA, repeated measures, mean ± SEM is shown). (**J**–**M**) The mitotic spindles in synchronized HeLa cells were visualized with α-tubulin (green), and the chromosomes—with CENP-B (kinetochores, red) and Hoechst 33342 (DNA, blue) (scale bars represent, 10 µm and apply for all panels). (**J**) Metaphase is considered defective when chromosomes (overlap of CENP-B and Hoechst stains) are not properly aligned (indicated with arrows) or damaged DNA (only Hoechst positive) is present. Example images of normal and defective metaphases are shown. (**K**) The percentage of defective metaphases from the total number of metaphases is plotted (600 metaphases per experiment, *n* = 4 independent synchronization experiments; Mann–Whitney *U-test*, min to max range with the median indicated as a line are shown). (**L**) Lagging chromosomes in anaphase are detected as CENP-B- and Hoechst-positive signal located in the middle part of the spindle (indicated with arrows). (**M**) The percentage of anaphases with lagging chromosomes from the total number of anaphases is plotted (400 anaphases per experiment, *n* = 4 independent synchronization experiments; Mann–Whitney *U-test*, min to max range with the median indicated as a line are shown). (SAC—spindle assembly checkpoint; CDC—cell division cycle protein; MAD2—mitotic arrest deficient 2; pVHL—von Hippel-Lindau protein; AURKB—Aurora kinase B; PLK1—Polo-like kinase 1; CENP-B—centromere protein B).

**Figure 3 cancers-12-02409-f003:**
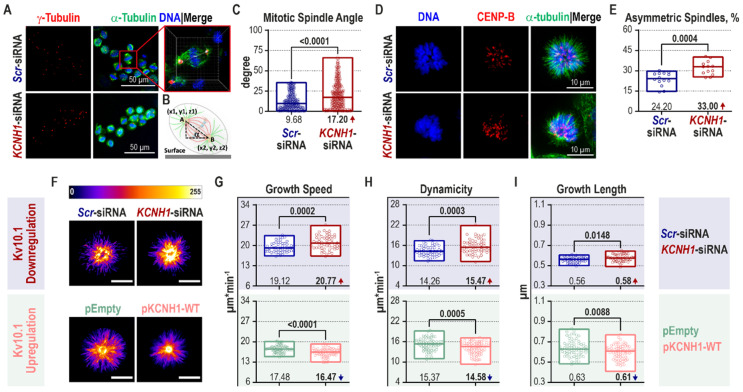
Tilted mitotic spindles and increased microtubule plus-end growth rates are observed in Kv10.1-deficient HeLa cells. (**A**–**C**) Hela cells were transiently transfected with either *Scr*- or *KCNH1*-siRNA and synchronized at the metaphase with consecutive thymidine and nocodazole treatments. (**A**) Maximal intensity projections of z-stacks (0.2 µm step size) of cells in metaphase are represented (scale bar, 50 µm) where the mitotic spindle is in green (α-tubulin), centrosomes—in red (γ-tubulin), and DNA—in blue (Hoechst 33342). The mitotic spindle angle was defined using the Imaris software as shown in the zoomed-in inset and schematically explained in (**B**). (**C**) The results of the mitotic spindle angle measurement are plotted (>50 events per experiment, *n* = 5 independent synchronization experiments; Mann–Whitney *U-test*, min to max range with the median indicated as a line are shown). (**D**–**E**) HeLa cells were transfected with respective siRNAs, grown until 70% confluency, treated with 2.5 µM DME for 4 h. The monopolar spindles were visualized with α-tubulin staining (green), and chromosomes with CENP-B (kinetochores, red) and DNA (Hoechst 33342, blue). (**D**) Example images of monopolar spindles are presented. (**E**) The percentage of asymmetric monopolar spindles is plotted (1500 monopolar spindles per condition per experiment, *n* = 3 independent synchronization experiments; Mann–Whitney *U-test*; min to max range with the median indicated as a line are shown). (**F**–**I**; Appendix A) HeLa cells were transiently co-transfected with pEB3-tdTomato and the indicated siRNAs (panel in light blue) or plasmid DNAs (panel in light green); 48 h later treated with 2.5 µM DME for 4 h. Microtubule growth was tracked in a series of images taken every 500 ms for 1 min. (**F**) Maximal intensity projections of time-lapse images over the recorded time are shown (“Red Fire” LUT is applied, scale bar, 10 µm). (**G**–**I**) Parameters calculated by the U-Track software are plotted (>25 cells per condition per experiment, *n* = 3 independent synchronization experiments; Mann–Whitney *U-test*, min to max range with the median indicated as a line are shown). (DME—dimethylenastron; LUT—look-up table; pEB3-tdTomato—a plasmid expressing end-binding protein 3 tandem dimer tomato fluorescent protein).

**Figure 4 cancers-12-02409-f004:**
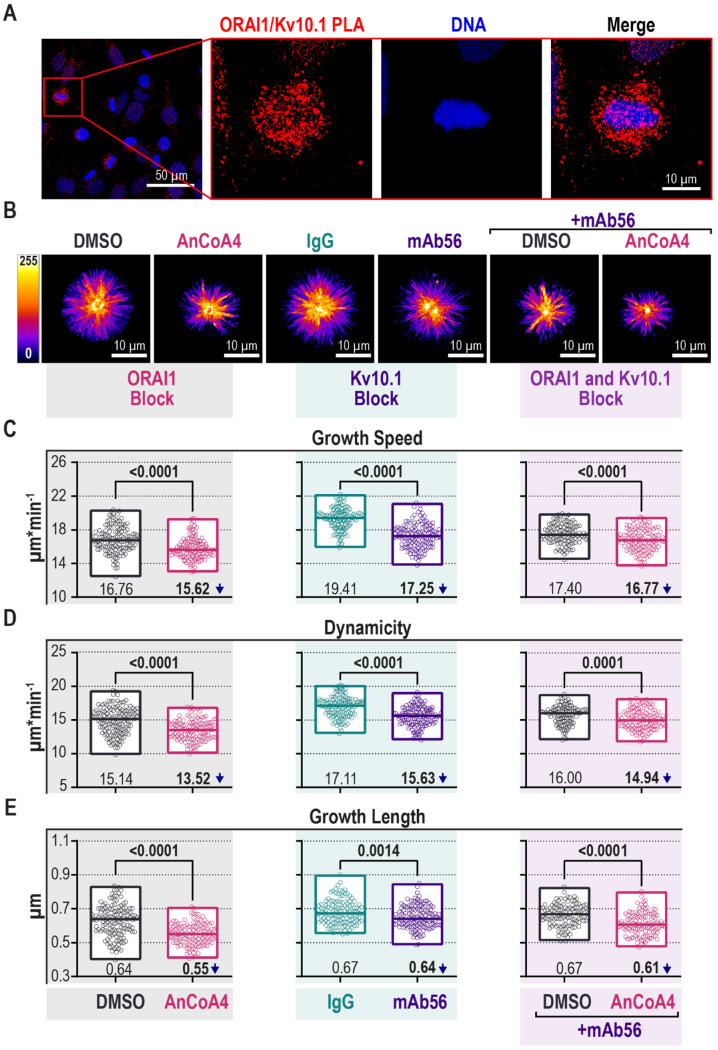
A block of either Kv10.1 or ORAI1 currents reduces MT dynamics in HeLa cells. (**A**) Example images of Kv10.1 and ORAI1 proximity ligation assay (red foci) in asynchronous HeLa cells (scale bar, 50 µm) and a mitotic cell (zoomed-in inset: scale bar, 10 µm; nuclei—blue) are given. (**B**–**E**; Appendix A) HeLa cells, transiently expressing pEB3-tdTomato were treated as follows: left grey panel—4h with the ORAI1 blocker AnCoA4 (10 µM) or empty vehicle (DMSO); middle green panel—24 h with the Kv10.1 blocker mAb56 or control IgG κ2b (both 5 µg/µL); right purple panel—a combination of both treatments. In all cases, the last 4 h of the treatments, the cells were exposed to 2.5 µM DME. Microtubule growth was tracked in a series of images taken every 500 ms for 1 min. (**B**) Maximal intensity projections of time-lapse images over the recorded time are shown (LUT is applied, scale bar, 10 µm). (**C**–**E**) Parameters of MT growth were defined with the U-Track software and plotted (>25 cells per condition per experiment, *n* = 3–5 independent synchronization experiments; Mann–Whitney *U-test***,** min to max range with the median indicated as a line are shown). (MT—microtubule; ORAI1—calcium release-activated calcium channel protein 1; DMSO—dimethyl sulfoxide; IgG—immunoglobulin G; mAb—monoclonal antibody).

**Figure 5 cancers-12-02409-f005:**
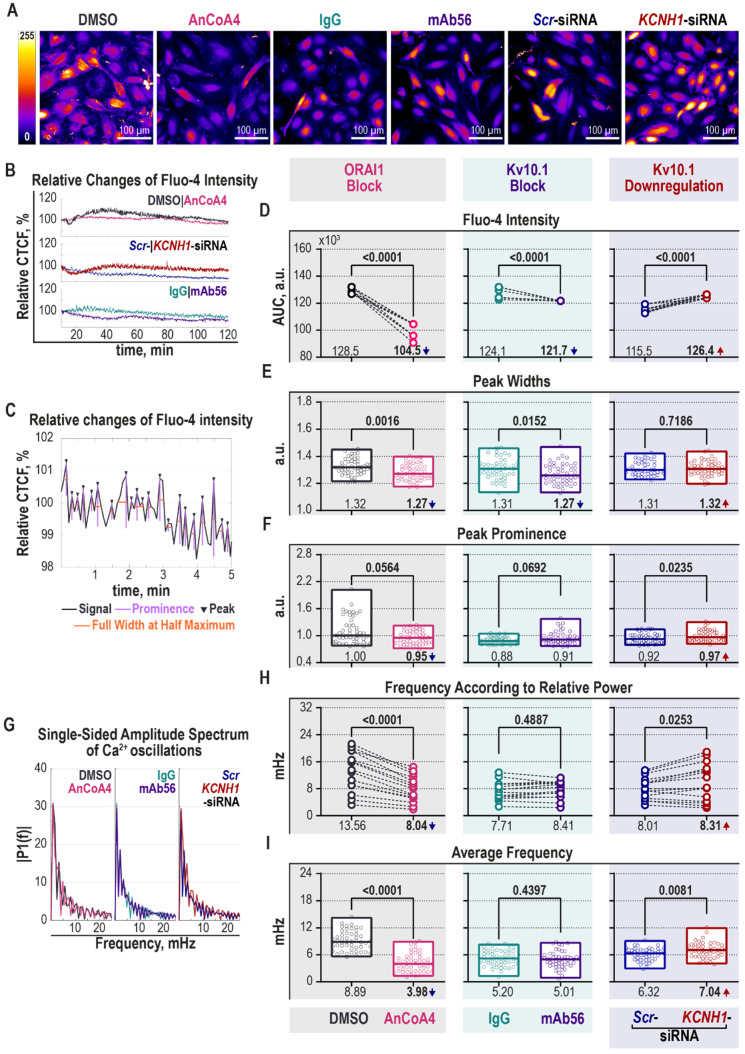
Kv10.1 regulates the intracellular [Ca^2+^] oscillations in HeLa cells during the G2/M phase. (Appendix A) Ca^2+^ imaging was performed at 7.5 h after initiation of the cell cycle from the G1/S border using HeLa cells preloaded with Fluo-4AM. Time-lapse images were taken every 5 s over 120 min, and changes in the intensity of Fluo-4 were recorded. (**A**) Maxima projection images of time-lapse series are shown (LUT is applied, scale bar, 100 µm). (**B**) The measured intensity was normalized for photobleaching, and the changes in Fluo-4 intensity relative to the corresponding time 0 are plotted over time. An example of such a normalized spectrum from one cell per condition is displayed. (**C**) An example of a normalized spectrum with identified local maxima (peaks, marked with triangles) from one cell is displayed where each identified peak is characterized by its height (prominence, shown in purple) and the full width at half maximum (shown in orange). (**D**–**F**) The total area under the recorded spectrum (**D**), the median prominence (**E**), and the median width at half maximum (**F**) were analyzed (50 cells per condition per experiment; *n* = 3 independent synchronization experiments; Mann–Whitney *U-test*, min to max range with the median indicated as a line are shown). (**G**–**I**) The changes in the intensity of Fluo-4 over time were converted into a frequency domain. (**G**) An example of a single-sided amplitude spectrum of relative changes of intracellular [Ca^2+^] defined by Fluo-4 from one cell per condition recording is shown. The frequencies were arranged in descending order according to the relative power, and the first 15 frequencies were selected for further analysis. (**H**) Distribution of the selected frequencies is given. (**I**) The average frequency was defined as an average of the product of the frequencies and the respective relative power (50 cells per condition per experiment; *n* = 3 independent synchronization experiments; Mann–Whitney *U-test*, min to max range with the median indicated as a line are shown).

**Figure 6 cancers-12-02409-f006:**
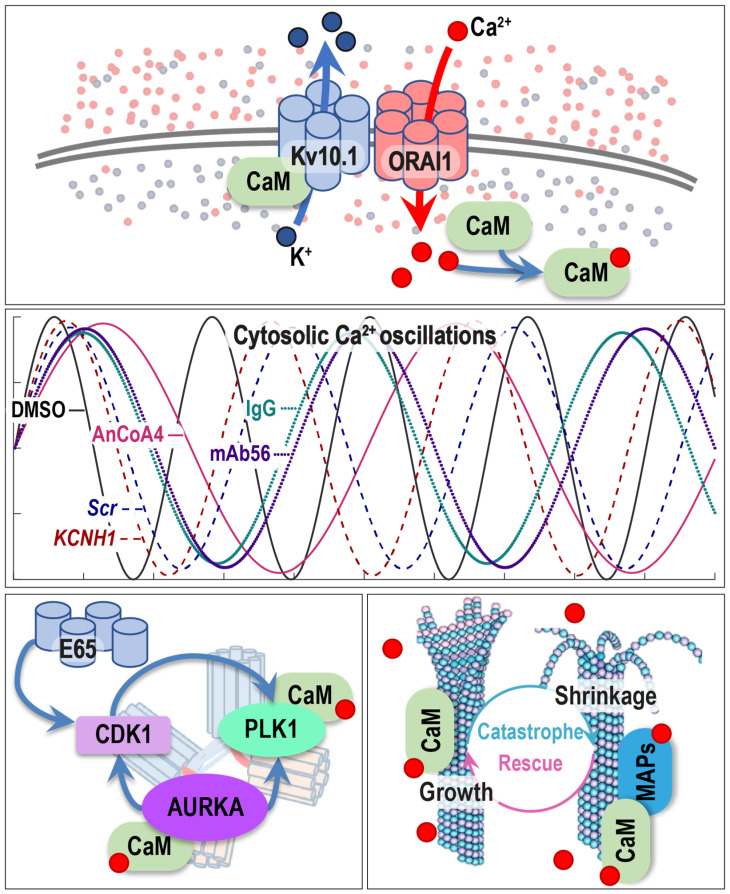
Working model of Kv10.1 action during the G2/M phase. Kv10.1 functionally interacts with ORAI1 creating a self-limiting circuit of Ca^2+^ entry (upper panel). Together, these two channels control the oscillations of cytosolic Ca^2+^: inhibition of any of the channels alters calcium entry (middle panel: simulation of calcium oscillations based on the empirically defined frequency values). Some proteins sense such oscillatory changes in cytosolic Ca^2+^. AURKA and PLK1 hold a high potential for such sensory proteins: both kinases are activated upon binding of Ca^2+^/CaM. Additionally, Kv10.1 split variant E65 directly activates CDK1 (low-left panel). As a result of the described changes, the MT assembly rate is modulated. Therefore, Kv10.1, through its canonic ion-channel function and protein-protein interactions, regulates the proper entry and progression through M phase (low-right panel). (K^+^—potassium ions; Ca^2+^—calcium ions; CaM—calmodulin; E65—split-form of Kv10.1; CDK1—cyclin-dependent kinase 1; PLK1—Polo-like kinase 1; AURKA—Aurora kinase A; MAPs—microtubule-associated proteins).

**Table 1 cancers-12-02409-t001:** Changes in the expression level of spindle assembly checkpoint (SAC) and DNA damage response (DDR) proteins upon Kv10.1 downregulation in synchronized HeLa cells.

Name	Checkpoint	Effect
p-ATR (Ser428)	DDR	Increased
p-CHK1 (Ser345)	DDR	Increased
CHK2	DDR	Increased
pVHL	SAC	Increased
AURKB	SAC	Decreased
PLK1	SAC	Decreased
MAD2	SAC	Increased
CDC20	SAC (APC/C)	Increased
WEE1	SAC (APC/C)	Increased
CDC27	APC/C	Decreased

**Table 2 cancers-12-02409-t002:** List of primary and secondary antibodies used for immunoblotting.

Antibody	Source	Dilution	Cat. Number	Manufacturer
p-ATR (Ser428)	Rb/pcl	1:1000	2853	Cell Signaling
p-BRCA1 (Ser1524)	Rb/pcl	1:1000	9009	Cell Signaling
p-CHK1 (Ser345) (133D3)	Rb/mcl	1:1000	2348	Cell Signaling
CHK1 (FL-476)	Rb/pcl	1:1000	sc-7898	Santa Cruz Biotechnology
p-CHK2 (Thr68)(C13C1)	Rb/mcl	1:1000	2197	Cell Signaling
CHK2 (DCS-270)	Ms/mcl	1:1000	sc-56296	Santa Cruz Biotechnology
p21 Waf1/Cip1	Ms/mcl	1:1000	05-345	Merck Millipore
p-histone H2AX (Ser139)	Rb/mcl	1:1000	sc-101696	Cell Signaling
CDC27 (AF3.1)	Ms/mcl	1:1000	sc-9972	Santa Cruz Biotechnology
CDC20	Rb/pcl	1:1000	ab26483	Abcam
AURKB/AIM1	Rb/pcl	1:1000	3094	Cell Signaling
PLK1	Ms/mcl	1:500	sc-17783	Santa Cruz Biotechnology
WEE1	MS/mcl	1:500	sc-5285	Santa Cruz Biotechnology
BUB3	Ms/mcl	1:1000	sc-376506	Santa Cruz Biotechnology
MAD2L1 (D8A7)	Rb/mcl	1:1000	4636	Cell Signaling
pVHL	Ms/mcl	1:200	556347	BD Pharmingen
Actin (C-11)	G/pcl	1:1000	sc-1615	Santa Cruz Biotechnology
Kv10.1	Rb/pcl	1:1500	9391	
Normal Mouse IgG	Ms	Isotype control	sc-2025	Santa Cruz Biotechnology
Normal Rabbit IgG	Rb	Isotype control	sc-3888	Santa Cruz Biotechnology
ECL Anti-Rabbit IgG, HRP Linked	D/pcl	1:10,000	NA934	GE Healthcare
ECL Anti-Mouse IgG, HRP Linked	Sh/pcl	1:10,000	NA931	GE Healthcare
Anti-Goat IgG (H + L)-HRP Conjugate	Rb/pcl	1:10,000	172-1034	BIO-RAD

MS—mouse; Rb—rabbit; G—goat; Sh—sheep; mcl—monoclonal; pcl—polyclonal.

**Table 3 cancers-12-02409-t003:** List of primary and secondary antibodies used for immunofluorescence.

Antibody	Source	Dilution	Cat. Number	Manufacturer
α-Tubulin	G/pcl	1:350	sc-7396	Santa Cruz Biotechnology
γ-Tubulin	Ms/mcl	1:700	sc-23948	Santa Cruz Biotechnology
CENP-B (H-65)	Rb/pcl	1:350	sc-22788	Santa Cruz Biotechnology
Alexa Fluor 488 anti-mouse IgG (H + L)	D/pcl	1:1000	A-21202	Invitrogen
Alexa Fluor 546 anti-rabbit IgG (H + L)	D/pcl	1:1000	A-11056	Invitrogen

MS—mouse; Rb—rabbit; G—goat; D—donkey; mcl—monoclonal; pcl—polyclonal.

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
