# Peer review of "Kv10.1 Regulates Microtubule Dynamics during Mitosis"

_cancers, 2020, doi:10.3390/cancers12092409_

Round 1

Reviewer 1 Report

The Authors of the submitted manuscript attempt to decipher role of Kv10.1 in microtubule dynamics during mitosis. As this protein in non-neural tissues almost exclusively expressed during in G2/M phase and is overexpressed in most of human tumour tissues the authors tried to assess its role mostly by silencing expression of the Kv10.1 encoding gene and inhibition of Kv10.1 and ORAI1 channels activity and also proximity ligation assay as well as advanced methods of assessing microtubule dynamics and spindle properties. They also used transient transfection and overexpression of Kv10.1.

They concluded that: 1. Kv10.1 decrease activates DNA damage checkpoint in the absence of DNA damage and to activation of SAC and decreased occurrence of lagging chromosomes in HeLa cells. 2. They show that Kv10.1 and ORAI1 co-localize in mitotic cells and blocking of both channels activity results in the reduction of MT dynamics. Their results suggest a common pathway affected by Kv10.1 and ORAI1. Finally the Authors document and discuss that Kv10.1 is involved in Ca2+ oscillation regulation and therefore in MT (spindle) dynamics.

The project is well designed and executed. Used methods and techniques are adequate and experiments are statistically assessed. I have no doubt that this manuscript deserves publication, however, before publication several point should be considered. In my opinion, the manuscript would be better suited to the general cell biology journal such as Cells. If however, Authors and Editors decide to consider it for Cancers, following issues should be considered and addressed before publication:

Major comment:

  1. It should be shown the efficacy of the Kv10.1 encoding gene knock-down via Western blotting. How was the statistics? How was the specificity of this KnD. How repeatable were the “transfections” from experiment to experiment as silencing of Kv10.1 encoding gene is considered? How about off-target effect of gene silencing.
  2. Blots in Fig.1 and in part in Fig.2 not always fit to the quantitation graphs.
  3. 5 A treatments have not been explained in details.

Author Response

The project is well designed and executed. Used methods and techniques are adequate and experiments are statistically assessed. I have no doubt that this manuscript deserves publication, however, before publication several point should be considered. In my opinion, the manuscript would be better suited to the general cell biology journal such as Cells. If however, Authors and Editors decide to consider it for Cancers, following issues should be considered and addressed before publication:

Thank you for your kind comments. Publishing it in a more cell biology-oriented journal was actually one of our alternatives. Nevertheless, we are persuaded that the field of ion channels in cancer requires to be exposed to the cancer field more directly. It is already over twenty years that we know that ion channels are a relevant agent in cancer, but this knowledge has been confined to ion channel specialists and physiologists. We, therefore, believe that Cancers is the right choice to reach scientists interested in all aspects of oncology.

Major comment:

  1. It should be shown the efficacy of the Kv10.1 encoding gene knock-down via Western blotting. How was the statistics? How was the specificity of this KnD. How repeatable were the “transfections” from experiment to experiment as silencing of Kv10.1 encoding gene is considered? How about off-target effect of gene silencing.

We thank the reviewer for pointing out this issue. We have now included a Supplementary Figure showing the efficacy of transient down- and overexpression. We have thoroughly validated the sequence used for silencing over the years. Additionally, we have systematically checked knockdown efficiency using western blot. To rule out non-specific effects, we used a non- targeting siRNA.

  1. Blots in Fig.1 and in part in Fig.2 not always fit to the quantitation graphs.

The slight differences between the presented WB images and the densitometry analysis can be explained by the fact that the quantification accounts all the experimental repeats and is normalized by the respective actin values.

  1. 5 A treatments have not been explained in details.

We included details on the treatment in the main text (lines 357 -359).

Reviewer 2 Report

The current paper aims to reconcile two different observations as it relates to Ca++ flux as induced by modulating KV10.1 and ORAI1. On one hand functional blockade and the other is knockdown. When opposite results are observed with CA++ flux using advanced methodology the level of proof that is required is greatly increased to provide a plausible mechanistic explanation. This due to the fact that it deals with a limited subject matter G2/M progression and MT spindle. This is difficult to resolve and clearly will require in future work to resolve these disparate results. 

The authors use multiple levels of evidence to justify the results found and if the results are correct and the data is validate we have to accept that those are the results although we do not understand fully why it is so. The view that a knock down by siRNA is complex since while affecting the KV10.1 through genomic changes it may also affect indirectly other parts of the cell since this is a cell line that has unstable chromosome makeup. On other hand, an antibody that targets the specific protein inhibition if effective can be almost complete. Therefore we may be dealing a direct versus less defined inhibition. 

The arguments regarding Ca++ AUC, oscillation is important to describe since it provides multiple levels of observations since only by amplitude the effect would have been minimal- not significant. The time line of exposure up to 120 minutes enabled a total exposure to testing. 

A table showing the different proteins that were decreased/increased  by the knockdown not only abbreviations would clarify the information- and suggested role (brief).

What is clearly missing is a conclusion where this data leads us since the study is done of cancer cell lines and not actually on primary tumors. So the data remains informative but descriptive.

The HeLa cells are far removed from human tumors. Therefore, whether KV10.1 inhibition that normally only present in neural cells is relevant for other tumors as a therapeutic agent is unanswered. Further which method should be used antibody, gene therapy other to achieve a therapeutic goal are not mentioned. Such information should be able to strengthen the paper. 

Author Response

A table showing the different proteins that were decreased/increased  by the knockdown not only abbreviations would clarify the information- and suggested role (brief).

Thank you for the suggestion. We have now included a table summarizing the effect of Kv10.1 KD on SAC and DDR protein level in the Results and Discussion section (209-211).

What is clearly missing is a conclusion where this data leads us since the study is done of cancer cell lines and not actually on primary tumors. So the data remains informative but descriptive.

The HeLa cells are far removed from human tumors. Therefore, whether KV10.1 inhibition that normally only present in neural cells is relevant for other tumors as a therapeutic agent is unanswered. Further which method should be used antibody, gene therapy other to achieve a therapeutic goal are not mentioned. Such information should be able to strengthen the paper. 

We thank the reviewer for his/her constructive comments. We are fully aware that work in cell lines cannot be directly translated to the clinical scenario. Nevertheless, we are persuaded that Kv10.1 will in the future be a relevant oncology target, but any clinical development of Kv10.1-directed therapies requires an understanding of the underlying molecular mechanisms (lines 649 – 656).

Reviewer 3 Report

In this manuscript the authors show that the voltage-gated potassium channel Kv10.1 regulates microtubule dynamics in mitosis. Kv10.1 modulates cytosolic calcium levels, through functional interaction with ORAI1, and inhibition of either Kv10.1 or ORAI1 stabilize microtubules. In contrast, Kv10.1 downregulation by siRNA increases mitotic microtubule dynamicity, causing spindle assembly checkpoint (SAC) activation and resulting in a decrease of lagging chromosomes.

The finding that Kv10.1, aberrantly expressed in a high percentage of tumors, might influence microtubule dynamics and, in turn, the SAC, a checkpoint essential to prevent aneuploidy, is relevant and of sure interest for the readers of Cancers. However, I believe some additional controls are necessary to definitively prove this point.

The different effect of Kv10.1 downregulation by siRNA and inhibition by mAb56 remains puzzling although, as the authors discuss, there are other examples in the literature. On one side, the authors should discuss if mAb56 has affinity for other Kv family members. On the other side, to clear up doubts, the authors should show the level of Kv10.1 downregulation obtained by siRNA in their experiments. In the material and methods section the authors state they have done this control by immunoprecipitation but they do not show any result. Likewise, the authors should also show the levels of overexpression of Kv10.1 they obtain by transfecting its cDNA in figure 3F-I. Moreover, to rule out any off-target effect, the authors should also demonstrate that the effects of an siRNA targeting a segment of the Kv10.1 3′ untranslated region are rescued by transfecting a full length Kv10.1-encoding plasmid.

The authors claim that their results in figure 2A-G reveal SAC activation in Kv10.1-depleted cells. To more convincingly demonstrate that, the authors should prove the presence of the MCC complex in their conditions, by showing that Mad2 e Cdc20 coimmunoprecipitate. Moreover, to make the interpretation of the mitotic protein levels more unequivocal, the authors should show that the time progression from the thymidine block release to anaphase is equal in the scrambled and Kv10.1 siRNA transfected cells by showing, for example, the levels of cyclin B or phospho-H3 proteins.

Minor points:

- The level of expression of the non-phosphorylated ATR protein are missing in figure 1/S1

- The references of the material and methods section are written in extenso and are not reported in the reference section

Author Response

The different effect of Kv10.1 downregulation by siRNA and inhibition by mAb56 remains puzzling although, as the authors discuss, there are other examples in the literature. On one side, the authors should discuss if mAb56 has affinity for other Kv family members.

Our previous work showed that mAb56 specifically inhibits Kv10.1 currents but not its closest homologue Kv10.2 (also known as hEAG2) and Kv11.1 (hERG) [1].

On the other side, to clear up doubts, the authors should show the level of Kv10.1 downregulation obtained by siRNA in their experiments. In the material and methods section the authors state they have done this control by immunoprecipitation but they do not show any result. Likewise, the authors should also show the levels of overexpression of Kv10.1 they obtain by transfecting its cDNA in figure 3F-I

We thank the reviewer for bringing up this issue. We have now included a Supplementary Figure in the “Immunoprecipitation” section of “Material and Methods” showing the efficacy of siRNA-mediated Kv10.1 downregulation and plasmid-DNA-driven overexpression of the channel (lines 660-664). 

Moreover, to rule out any off-target effect, the authors should also demonstrate that the effects of an siRNA targeting a segment of the Kv10.1 3′ untranslated region are rescued by transfecting a full length Kv10.1-encoding plasmid.

We agree with the reviewer that 3’ UTR-targeted siRNA is less prone to off-target effects and compatible with rescue experiments. However, during the process of selection of the siRNA sequence that we have been using for many years, we were not successful in obtaining strong downregulation of Kv10.1 message using non-coding target sequences. Nevertheless, the fact that Kv10.1 overexpression shows and effect opposite to knockdown supports strongly the specificity of the effects described in the manuscript.

The authors claim that their results in figure 2A-G reveal SAC activation in Kv10.1-depleted cells. To more convincingly demonstrate that, the authors should prove the presence of the MCC complex in their conditions, by showing that Mad2 e Cdc20 coimmunoprecipitate.

We thank the reviewer for her/his comment. In fact, we had overlooked additional evidence for SAC activation already presented in our previous work, since cyclin A2 degradation works as a readout of SAC activity [2], and we had already reported a delay in cyclin A2 degradation upon Kv10.1 knockdown [3]. We have now corrected this omission in lines 139-142. We still agree that a direct demonstration of the presence of the MCC complex would be very compelling evidence for SAC activation. We do not have the possibility to produce such data in the ten days allowed for the revision, but nevertheless we believe that the central message of the manuscript is not SAC activation upon Kv10.1 knockdown, but the molecular mechanism underlying the delayed cell cycle progression, which likely includes –but is not reduced to– SAC activation.

Moreover, to make the interpretation of the mitotic protein levels more unequivocal, the authors should show that the time progression from the thymidine block release to anaphase is equal in the scrambled and Kv10.1 siRNA transfected cells by showing, for example, the levels of cyclin B or phospho-H3 proteins.

We thank the reviewer for this comment. Our previous work showed that cell cycle progression is affected by Kv10.1 knockdown. This was documented by quantifying the levels of cyclin B (kindly see Figures 1 and 5 in [3]). Nevertheless, our microtubule dynamics experiments were all performed in cells already in mitosis.

Minor points:

- The level of expression of the non-phosphorylated ATR protein are missing in figure 1/S1

Unfortunately, we failed to detect total non-phosphorylated ATR via immunoblot analysis. Nevertheless, comparing the active phosphorylated form of ATR between the described conditions is more relevant when assessing the activation status of DNA damage-activated checkpoint.

- The references of the material and methods section are written in extenso and are not reported in the reference section

Thank you for noticing this. Now, we have corrected the style of the references also in the Material and Methods section and included in the bibliography.

  1. Gomez-Varela, D.; Zwick-Wallasch, E.; Knotgen, H.; Sanchez, A.; Hettmann, T.; Ossipov, D.; Weseloh, R.; Contreras-Jurado, C.; Rothe, M.; Stuhmer, W., et al. Monoclonal antibody blockade of the human Eag1 potassium channel function exerts antitumor activity. Cancer Res 2007, 67, 7343-7349, doi:10.1158/0008-5472.CAN-07-0107.
  2. Collin, P.; Nashchekina, O.; Walker, R.; Pines, J. The spindle assembly checkpoint works like a rheostat rather than a toggle switch. Nat Cell Biol 2013, 15, 1378-1385, doi:10.1038/ncb2855.
  3. Urrego, D.; Movsisyan, N.; Ufartes, R.; Pardo, L.A. Periodic expression of Kv10.1 driven by pRb/E2F1 contributes to G2/M progression of cancer and non-transformed cells. Cell Cycle 2016, 15, 799-811, doi:10.1080/15384101.2016.1138187.

Round 2

Reviewer 3 Report

The authors have addressed, or at least satisfactorily discussed, all my concerns and I thank them for that. I have no further request and I think this manuscript is now acceptable for publication.